# International Coordination of Research Ethics Review: An Adequacy Model

Adrian Thorogood [1,2,]* and Michael J. S. Beauvais [1]

[1] Centre of Genomics and Policy, McGill University, Montréal, QC H3A 0G4, Canada; michael.beauvais@mail.mcgill.ca
[2] Bioinformatics Core, Luxembourg Centre for Systems Biomedicine, University of Luxembourg, 4365 Esch-sur-Alzette, Luxembourg
[*] Correspondence: adrian.thorogood@uni.lu

**Abstract:** International direct-to-participant (DTP) genomics research involves the use of mobile technology to recruit, consent, and study participants remotely. This model can facilitate research across broad geographies and many countries, but must also comply with the norms of multiple recruitment jurisdictions, with each jurisdiction typically requiring at least one local research ethics review. Each additional research ethics review increases bureaucratic hurdles without necessarily strengthening the protection of participants' rights and interests. For DTP genomic research, obtaining a review may in fact be impossible in the absence of a local research partner. This paper proposes an "adequacy" approach, inspired by data protection law, to coordinate the regulation and oversight of international DTP genomics research. This involves one country voluntarily assessing whether another country's research ethics reviews are equivalent to its own, in terms of objectives and effectiveness. Ethics-approved projects led by researchers from countries recognized as adequate are deemed to comply with local norms, eliminating the need for a duplicative local review. Adequacy preserves the sovereignty of countries to determine their own regulatory aims and which other countries to trust. It therefore provides a voluntary, incremental path towards greater global coordination of health research oversight.

**Keywords:** research ethics; direct-to-participant; international collaboration; genomics; research ethics committee; law

## 1. Introduction

Direct-to-participant (DTP) genomics research allows researchers to recruit, consent, interact with, and collect samples and data from participants remotely. Recruitment and engagement can be conducted through social media. Electronic consent can be obtained using mobile apps. Do-it-yourself sample kits and mobile health survey and sensor data can be contributed by mail or over the internet. These tools have the potential to liberate health research from the confines of institutional walls. DTP genomic research could allow research to rapidly scale up internationally, without a corresponding need to establish a research presence or partner in each recruiting institution or country. DTP genomic research may be especially effective to study rare diseases where patient numbers are too low to conduct research within a single institution or country, or to justify the overhead of international, multi-centre research collaborations [1].

Health research projects with human participants must generally be approved and supervised by an institutional review board (IRB). IRBs review the ethical acceptability of research involving human participants, with a principal mandate to protect participants from harm [2]. International research projects are typically subject to oversight from multiple IRBs—one at each participating site or country. In a 31-country survey of ethico-legal experts (hereafter: the "Country Reports"), Rothstein et al. found that international DTP genomic research predominantly requires ethics reviews in both the researcher's

country, as well as in each country where participants are recruited [1,3]. This survey explored requirements applying to international DTP genomic research, including IRB review, informed consent, local collaborators, restrictions on cross-border sample and data transfers, and rules specific to human genetic research. The survey also explored broader regulatory attitudes in the country regarding direct-to-consumer genetic testing, economic protectionism, and bio-expropriation. A central finding of the study was that most countries require a local IRB review of foreign-led DTP genomic research projects. This duplicative oversight is potentially unnecessary, increasing paperwork and delays without necessarily strengthening protections for human subjects or research validity. This problem is already being addressed for multi-site research within countries who have adopted mechanisms to coordinate IRB review [4,5]. For international research, however, no coordination mechanisms for IRB oversight exist.

The requirement to obtain an IRB approval in each country where research is conducted may act as a de facto prohibition on international DTP genomics research. Without a local collaborator or recruiting site, it may be impossible for a foreign-led DTP project to obtain a local IRB approval.

To address this problem, Rothstein et al. have proposed that the duplicative ethics review of international DTP genomics research can be eliminated by an "adequacy" approach [1]. Adequacy is a concept in European Union (EU) data protection law, which permits transfers of personal data outside of the EU to countries that can demonstrate that their norms provide essentially the same protection as EU norms. Data protection norms are increasingly relevant for global, data-intensive scientific research, but they are not the direct focus of our inquiry [6]. Rothstein et al.'s proposal is that an adequacy model might be transposed from data protection law and applied in the field of research ethics, in order to eliminate duplicative IRB oversight of international DTP genomics research in a principled and practical way.

Our paper builds on Rothstein et al.'s proposal by analysing the conceptual and practical aspects of applying an adequacy model to health research regulation. First, we highlight that the real challenge is not coordination between IRBs overseeing a particular project per se, but rather coordination between national regulatory frameworks. International health research projects must comply with multiple sets of health research norms from multiple jurisdictions. An IRB review in each country is one way to ensure that the design and conduct of the research complies with the health research norms of each country. The requirement, however, can be expensive and time-consuming. Given the commonalities between most countries' health research norms, the duplication may also do little to improve the protection of human subjects.

Conceptually, adequacy involves an international comparative law methodology, with a focus on function. Researchers can only hope to have their IRB approvals recognized in another country if the health research norms of both countries are demonstrated to be essentially equivalent, both in terms of their purpose and their effectiveness. The health research norms of the two countries must share common, core purposes, and an ability to effectively attain those purposes. We examine in detail how this equivalence of regulatory purposes and effectiveness might be demonstrated in the domain of research ethics. We also consider the practical advantages and challenges to research ethics adequacy. The primary advantages of research ethics adequacy are that it eliminates duplicative research ethics oversight while still ensuring compliance with multiple countries' standards. As adequacy decisions are unilateral, adequacy also allows countries to uphold their standards of participant protection, while still incrementally facilitating international DTP genomic research on a country-by-country basis. The primary challenge of adequacy is that reaping its rewards requires countries to act both altruistically—to facilitate science led by other countries—and collectively—as benefits will only be realized by widespread, mutual recognition. Despite these shortcomings, we conclude that adequacy presents a cautious and therefore viable way forward to improve the global coordination of health research regulation and oversight.

## 2. Adapting Data Protection Adequacy to Research Ethics Review

This section introduces the concept of adequacy in data protection law and explains how it might be exported to apply in the field of research ethics. Exporting this concept is not without precedent. The European *Data Governance Act* is exploring a similar approach for protecting the international transfers of intellectual property rights [7].

### 2.1. Data Protection Adequacy

Data protection norms have evolved alongside the increasingly international nature of telecommunications technologies and data processing activities [8]. Data protection aims to ensure that the personal data of individuals are processed in a manner that is lawful, transparent, fair, and secure. Ensuring that a sufficiently rigorous level of data protection is achieved in the face of international transfers of data, which introduce the risk that otherwise unlawful data processing happens elsewhere, has led to a variety of regulatory approaches [9,10]. One way of ensuring that regulatory objectives are sufficiently attained for international transfers under the *General Data Protection Regulation* (GDPR) has been through the issuance of an "adequacy decision" by the European Commission, the executive branch of the EU (Art 45) [11]. The issuance of an adequacy decision recognizes that the recipient country's data protection regime offers an "essentially equivalent" level of data protection, such that complying with foreign norms achieves a similar outcome as though the provisions of the GDPR were followed [12]. In effect, an adequacy decision represents trust in foreign norms and oversight.

An adequacy approach analyses a foreign normative framework in its entirety, rather than granularly analysing how foreign norms govern a specific data processing activity. An adequacy assessment is a functional comparison, looking at regulatory purposes and effectiveness. The form of data protection law is not determinative, as regulatory objectives can be achieved through other means [13]. For example, the EU recognizes that a company processing personal data in Argentina according to Argentinian data protection law offers individuals an essentially equivalent level of protection as EU law [14]. Companies in Europe may thus freely send data to Argentina-based recipients without worry of breaching their own obligations under EU law to protect personal data. The Argentinian company simply complies with Argentinian data protection law and is thus deemed to offer an essentially equivalent level of data protection. How might this adequacy model be adapted to coordinate regulation and oversight of DTC genomics research?

### 2.2. Adequacy in the Research Ethics Context

2.2.1. Step One: Define Central Objectives of Local Norms and Oversight

A research ethics adequacy model would permit foreign researchers from countries recognized as adequate to conduct DTP genomics research with only their foreign IRB approval. Adequacy requires the search for functional equivalents between countries' norms. By norms, we mean a rule that serves as a standard against which conduct may be measured. Depending on the country, norms may have as their source statutes, regulations, bioethical frameworks, or other normative documents. Functional equivalence is a concept from comparative law referring to a situation where an institution or norm in legal system B performs an equivalent function to the norm in legal system A [15]. The analysis may also be problem-focused by asking "[w]hat solutions does the normative system provide for the problem X?" The sharing of problems across countries allows for the comparability of norms [16]. Adequacy involves a comparison of core purposes—corresponding to societal concerns—and institutional effectiveness. Core purposes may be expressed as principles that serve as the foundation for more specific norms. In order to function, an adequacy approach must focus on a comparison of core purposes, which may need to be distinguished from peripheral purposes or detailed rules. Indeed, the greater the number and detail of the core purposes compared, the less likely it is that foreign normative frameworks will be recognized as adequate. There is no pressure, however, for all countries to align on the same purposes. An adequacy model is primarily procedural; it does not dictate

a universal set of purposes that a country's regulatory framework should achieve. Each country maintains its sovereignty to define its own regulatory framework and adequacy referential. As adequacy decisions are unilateral, countries also maintain their freedom over which other countries they recognize as adequate.

To implement an adequacy model, countries must first look inwards and define the core purposes of their own normative framework and the institutions that fulfil them. In the context of EU data protection, the European Data Protection Board has established an Adequacy Referential, which specifies the core elements of the EU's data protection regime, against which a foreign regime is to be evaluated for determining adequacy under the GDPR [13]. The referential includes core concerns of data protection, such as the purpose limitation principle and the need for personal data to be processed pursuant to a lawful basis, as well as procedural mechanisms such as oversight processes.

For research ethics, a national bioethics committee or analogous body would have to develop an analogous referential for research ethics norms. One could start by articulating high level principles such as the Belmont Report's famous tripartite principles of respect for persons (autonomy), beneficence, and justice [17]. These core principles guide the development of more specific rules or standards. Two common sets of rules include ensuring informed participant consent and robust research ethics review. An adequacy referential for research ethics may be likely to include (but not necessarily be limited to) the following elements:

- Freely given, informed consent
  - Norms relating to the informed quality of consent
    - Information-giving obligations
    - Plain-language requirements for consent materials
    - Specificity of consent
  - Norms relating to the freely given quality of consent
    - Compensation and undue influence
    - Age of consent (minors)
    - Legally authorized representatives (decisionally vulnerable adults)
- Protocol review
  - Participant interests
    - Participant protection, e.g., risk–benefit ratio
    - Promotion of participant welfare, e.g., return of individual results
  - Enabling conditions for protocol review
    - REC mandate
    - REC composition
    - REC transparency and accountability

Comparative approaches to research ethics review already exist in the form of initiatives such as the Global Alliance for Genomics and Health's Ethics Review Recognition Policy [18]. The policy includes common elements of research ethics review such as the need for a research protocol, controls against conflicts of interests, an analysis of risks and benefits flowing of participation, and considerations for vulnerable populations, etc.

### 2.2.2. Step 2: Assess if Foreign Normative Framework Is Functionally Equivalent

The adequacy reference is then used to assess whether foreign normative frameworks are functionally equivalent in terms of purposes and effectiveness, and—where successful—to issue an adequacy decision. The national bioethics committee undertaking research ethics adequacy must look for functional equivalents within foreign research norms [19]. Identifying functional equivalents requires the national bio-ethics committee to use the objectives identified in the first step to examine foreign norms.

There is arguably a high level of international harmonization regarding the objectives of national research ethics frameworks, which should give cause for optimism that an

adequacy model would be practically feasible. National frameworks share common provenance in international documents such as the World Medical Association's Declaration of Helsinki [20] and the Council for International Organizations of Medical Sciences and World Health Organization's International Ethical Guidelines for Health-related Research Involving Humans [21], as well as highly influential American frameworks, as that country drove international-scale research projects [22,23]. The Country Reports confirm many underlying similarities between national frameworks. Presumably, some countries have adopted frameworks that address specific local concerns, such as research with indigenous populations. Others may have updated their frameworks at a greater speed, or simply drifted apart over time. Countries are of course free to establish their own adequacy referential, to define core regulatory purposes, and to unilaterally determine whether foreign frameworks are equivalent.

Greater challenges may arise when assessing the equivalent effectiveness of two national frameworks. There is already a rich literature looking at research ethics capacity in developing countries that highlights the importance of systems over formal written norms [24]. Empirically verifying the accountability and successes of IRBs is a difficult question that lacks agreed-upon criteria, as well as a central repository for relevant data [25–27]. National efforts to establish standards, certification, and auditing of IRBs could assist in providing such assurance. Reviewing a sample of actual IRB decisions from the country in question may also be informative in gauging the most important concerns for human genomic research projects.

Comparative law involves many interpretive challenges. First, an adequacy model must define a threshold for meeting functional equivalence. The applicable standard may indeed vary from one country to another, or even in time. In EU data protection, an "adequate" level of data protection has evolved such that an "adequate" level of data protection means an "essentially equivalent" level of data protection [12]. This change has meant that that normative frameworks are examined with greater scrutiny. An examination of the European Commission's adequacy decisions over time shows that pre-Schrems decisions were less exigent (e.g., Canada, Uruguay) than post-Schrems (e.g., Japan, UK) [28]. The more exacting the standard that informs a determination of adequacy, the closer the analysis begins to approximate a point-by-point analysis, the consequence of which is a reduced chance of a determination of adequacy. Fundamental to an adequacy approach is the fact that each country remains free to set its own regulatory objectives, to decide on the level of detail of its referential, and to decide what other countries have equivalent ethics review. Thus, adequacy does not ask countries to compromise local standards for protecting human subjects. It does, however, rely on a genuine desire to promote international research for the common good.

Recognizing adequacy in research ethics may in fact be less risky than doing so in data protection. Foreign-led international DTP research would still be reviewed by a foreign IRB, which may guard against insistence on near-exact copies of local norms. Indeed, data protection norms tend to have few ex ante checks, unlike research with human participants [29].

Adequacy also requires that foreign norms be accurately interpreted and understood in relation to their objectives. Countries that are interested in being the subject of an adequacy decision can facilitate this process by publishing their normative frameworks in English, or the working language of whichever country from which they would like to be deemed adequate. Additional steps could include providing a comprehensive, detailed explanation of the norms and their objectives so that a proper comparison may be made. If needed, the national bioethics committee or legislator generating norms could also make itself available to respond to queries from their foreign equivalents.

### 2.2.3. Step 3 Issuing an Adequacy Decision: Carve Outs and Conditions

An adequacy decision is essentially a decree that research projects supervised by an IRB in a foreign country are considered equivalent to projects supervised by a local IRB.

Given adequacy's emphasis on enabling conditions such as oversight and enforcement mechanisms, in principle, only protocols reviewed by IRBs with appropriate oversight are able to avail themselves to an adequacy decision. We envisage adequacy as an approach for researchers based at academic or governmental institutions whose IRBs are subject to oversight mechanisms as required by law (as is common in Europe) or as a condition of public funding (as is common in North America).

A foreign normative framework may be generally equivalent, while failing to take into account a specific local concern. One workaround to this problem is to exceptionally carve out certain types of research from the ambit of the adequacy decision, and to consequently require review by a local IRB. Concerns related to research involving indigenous groups, for example, are unlikely to be adequately handled through recourse to foreign norms. Indeed, as a response to the interplay of epistemologies presented by genomic research with indigenous groups, there is a recognized need for norms specific to this area of research. These range from specific chapters in national bioethical frameworks [30] to the creation of separate frameworks and procedures specific to genomic research with indigenous groups [31,32]. Issues such as data sharing may further present distinctive concerns where the data of indigenous communities are implicated [33]. The history and development of such norms are highly contextual and are the result of sustained engagement between indigenous communities and the research and bioethics communities in a particular place.

Another approach where a specific local concern is overlooked is for the adequacy decision to impose additional conditions. In data protection, for example, the European Commission found Japan's laws insufficiently protected from certain types of special category data, but this was addressed by the adequacy decision requiring Japanese companies to take additional measures [34]. A similar approach could be taken for research ethics adequacy by providing supplemental rules for foreign-led research. Limits and conditions on adequacy should be tied to core regulatory purposes, however, as they can undermine adequacy's efficiencies by introducing interpretive complexity.

## 3. Research Ethics Adequacy: Practical Considerations

### 3.1. Advantages

Adequacy's central advantage is eliminating the need for duplicative IRB approval and oversight. Normally, researchers conducting international DTP genomics research would need IRB approval in both their own country and in each country where they recruit. The researcher's IRB is likely to insist that the project obtain foreign IRB approval(s) to demonstrate compliance with foreign research norms. With an adequacy decision, researchers would only have to demonstrate compliance with their own country's norms, and they would be deemed compliant with the norms applicable in the other country. Adequacy recognizes equivalency between regulatory frameworks; there is a need to check that each project is compliant with foreign norms. At most, researchers would have to demonstrate that their research project met any applicable limitations and conditions (see above). Streamlined oversight would facilitate research, especially for understudied conditions, and accelerate advancements in knowledge and human health.

Adequacy simultaneously preserves local standards of human subjects' protections and sovereignty. The effectiveness of local regulations is an important concern for all sovereign countries, as evidenced by the common requirement of local IRB review of foreign-led research (see Country Reports). A country is free to generate its own robust standard of essential equivalence, and to determine who meets it, as adequacy decisions are unilateral. Adequacy decisions provide flexibility to limit types of foreign-led research that raise specific local concerns, or impose conditions. Adequacy is recognized incrementally, one trusted country at a time, giving countries time to build trust in their partners and confidence in the value of greater research collaboration. In terms of duration, adequacy decisions can be subject to expiry or review after a certain period or after a major regulatory change. Monitoring and reporting obligations, such as those included in the European Commission's adequacy decision for the United Kingdom, can ensure an ongoing assess-

ment of regulatory purposes and effectiveness [35]. In short, adequacy maintains notions of sovereignty but re-configures them with international collaboration in mind.

Finally, adequacy approaches may promote greater information exchange, community building, and coordination between health research regulators internationally [36]. By engaging in the process of articulating the essential aspects of a country's normative framework, an adequacy approach may inform policymakers about the essential ways the normative frameworks in their countries differ from those of other countries. Policy makers may start to pay closer attention to the compatibility of their regulations with those of other countries. The recognition process also presents an opportunity for direct exchange, mutual learning, and collaboration between regulators. In this way, adequacy approaches may provide the first steppingstones towards multilateral solutions, such as mutual recognition treaties.

### 3.2. Disadvantages

Adequacy assessments can be highly bureaucratic and expensive. Adhering to the analytic method of adequacy may be time-consuming and require additional resources and skills for national bioethics bodies. Conversely, strong incentives may be lacking. The primary beneficiaries of adequacy recognition are researchers in the foreign country. Recognition may even disadvantage researchers in the participant's country, by reducing incentives for foreign researchers to work with local collaborators. There may be some hidden benefits of duplicative ethics review that are lost, such as the opportunity for low-income countries to develop local ethics review capacity [37]. Benefits to the participant's country will at best be indirect, such as improved knowledge or treatments available on the world market, and, for poorer countries, such benefits may never truly materialize. Global inequities in access to medicines affect many countries, resulting largely from the high costs of medicines, tied to intellectual property rights and resulting monopolies [38]. This situation may discourage low- and middle-income countries from streamlining an ethics review of foreign research. For countries seeking adequacy recognition, incentives may also be modest and costs non-trivial. Adequacy assessments are unilateral and may impose conditions on foreign countries to implement regulatory changes, without negotiation or mutual accommodation. At least for similarly situated countries, reciprocal adequacy decisions can be a way to frame adequacy as providing mutual benefits. Adequacy approaches should also be promoted as fundamentally aiming to advance the public good in advancing science and improving care for all, rather than narrow quid pro quos.

Adequacy decisions can also sometimes be politicized, as has sometimes been the case under the EU's data protection adequacy regime [39]. As to the political dynamics of adequacy, concerns of autonomy and sovereignty may rise to the surface and prevent cooperation. An attempt to establish a joint USA–Kenya IRB, for example, was frustrated because Kenyan officials saw such an endeavor as an impingement on the ability of Kenyan institutions to review research on their own [40,41] Admittedly, the stakes are slightly lower in the context of health research as opposed to global data flows, with the emphasis being more on scientific advancement rather than commercial gain.

Adequacy is also an exceptional regime, meaning that cross-border activity is closed by default. Indeed, the EU adequacy regime for data protection has only recognized 14 jurisdictions as adequate [42], and restricts transfers to all other countries, unless some other type of potentially complex mechanism is used (Arts 46–48) [11]. The Country Reports suggest that at least some countries do not regulate (and, at least technically, do not prohibit) foreign-led DTP genomics research. For these countries, an adequacy approach would actually represent a tightening, rather than a loosening, of rules. However, if these countries are consciously open to foreign-led research (and not simply failing to regulate such activity), they can always adopt an adequacy regime that would provide greater predictability for foreign researchers.

Adequacy poses collective action problems. A critical mass of countries would need to adopt adequacy regimes and issue adequacy decisions to have a meaningful effect on

facilitating international DTP genomics research. For example, if Peru declares 20 countries to be adequate, researchers in those countries can still only recruit in Peru. They still need ethics approvals or collaborators in any third country. Many adequacy regimes and many adequacy decisions are needed to meaningfully facilitate international research. This collective action problem could be allayed if countries with high rates of research outputs and strong participant protections act as first movers through the issuance of adequacy decisions [1]. If, for example, the USA were to say that researchers in Canada, the UK, France, and Singapore following their local research norms for DTP research would be adequate under the *Common Rule*, it is likely that in the spirit of mutual recognition, reciprocal decisions could be adopted in favour of the USA [43]. Adequacy decisions must also be compatible; different limitations and conditions would lead to an unworkable patchwork. A voluntary framework for adequacy could assist with this problem. The framework could include standard carve outs such as research with indigenous communities, analogous to standard prudential carve-outs in international trade agreements covering financial services, which allows countries to adopt otherwise prohibited discriminatory measures (regulations) to maintain the stability of their financial systems [44].

Even a cautious opening of borders to foreign researchers is not without costs, or risks to sovereignty, competitiveness, and equity. We do not envisage adequacy as a panacea for global research. Some countries may decide that their local norms are fundamentally incompatible with those of other countries. However, adequacy may hold great promise for groups of like-minded countries, who are willing to prioritize the advancement of knowledge, and the well-being of patients locally and around the world.

The advantages and disadvantages of an adequacy model must be considered in light of a lack of viable alternatives. The status quo of requiring local IRB oversight of foreign-led projects imposes a potentially unnecessary burden. International DTP genomics research was not foreseen by regulations, perhaps under the assumption that all international research would establish a local site and collaborator. As such, the status quo may effectively—and perhaps unintentionally—prohibit foreign-led DTP projects. Countries could simply choose not to regulate foreign-led DTP genomics research at all, though this would be a radical departure from notions of sovereignty and accountability. Despite some methodological and practical drawbacks, adequacy presents a practical and incremental path forward.

## 4. Conclusions

Increasing connectivity is only making health research more global. Regulation and oversight of health research, however, remains a stubbornly local affair. This mismatch may delay or even frustrate research projects, without necessarily improving the protection of human subjects or the ethical acceptability of research. Countries have an ethical obligation to further both the benefits of scientific advancement and the protection of participants by coordinating health research regulation and oversight internationally. Adequacy presents a promising way forward, maintaining local human subjects' protection standards and sovereignty, while eliminating duplicative IRB review and multi-jurisdiction compliance assessments. Country-by-country recognition allows countries to open incrementally to foreign researchers on their own terms. Where necessary to protect particular local interests, such as the sovereignty of indigenous groups, the scope of adequacy decisions can be subject to limitations or conditions. The resulting international exchanges may lay the groundwork for greater regulatory harmonization and coordination.

**Author Contributions:** The authors contributed equally to all aspects of the manuscript, including conceptualization, methodology, writing the original draft, and review and editing. All authors have read and agreed to the published version of the manuscript.

**Funding:** This research was funded by the National Human Genome Research Institute of the US National Institutes of Health, grant No. 5R01HG009914-02, Regulation of International Direct-to-Participant Genomic Research, Mark A. Rothstein and Bartha Maria Knoppers, PIs.

**Institutional Review Board Statement:** Not applicable.

**Informed Consent Statement:** Not applicable.

**Data Availability Statement:** Not applicable.

**Acknowledgments:** The authors thank Bartha Maria Knoppers and Mark A. Rothstein for comments on an earlier version of this manuscript. All errors and omissions are the authors' own.

**Conflicts of Interest:** The authors have no conflict of interest to declare.

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
