# Peer review of "International Coordination of Research Ethics Review: An Adequacy Model"

_philosophies, doi:10.3390/philosophies6040093_

Round 1

Reviewer 1 Report

This is a well written and argued paper on using adequacy determinations for review of multi-national human subjects research.  It builds on the work on Rothstein et al (2019), develops a 3-step process for adequacy determinations, and considerations advantages and disadvantages the approach.  I have no suggestions for revision.

Author Response

Reviewer 1 had no suggestions for revision. 

Reviewer 2 Report

I read with interest the manuscript on the theme: "International coordination of research ethics review: an adequacy model" I think it could be an interesting document for readers and the general public. The problem of sample selection and research through selection methodologies with electronic devices is a great and innovative topic on which it is necessary to question. The authors suggest an adequacy model that may be interesting, the authors give a balanced analysis as they also state the disadvantages. In general the article is worthy of publication however I think it is important to strengthen some aspects:
- one of the fundamental concepts that should be respected in the selections is certainly the suitability but in the current context it would be possible to suggest that fairness is also important. In other very current contexts the importance of fairness has been emphasized, eg. DOI: 10.3390/vaccines9060538  the authors could briefly refer to this.
- the authors clearly show the disadvantages of the analyzed method but could they add something about the possibilities of overcoming these disadvantages?
i think these brief changes could improve a good manuscript

Author Response

In response to reviewer 2's comments, we have developed some arguments concerning fairness and developmental disparities in the context of the adequacy model (see section 3.2 - lines 305-340). Specifically, we highlight the intersection of health research ethics and access to medicines in low- and middle-income countries. We further engage with ways to overcome disadvantages.

Round 2

Reviewer 2 Report

in my opinion the article is publishable